# Carotenoid Biosynthetic Genes in Cabbage: Genome-Wide Identification, Evolution, and Expression Analysis

**DOI:** 10.3390/genes12122027

**Published:** 2021-12-20

**Authors:** Wenxue Cao, Peng Wang, Limei Yang, Zhiyuan Fang, Yangyong Zhang, Mu Zhuang, Honghao Lv, Yong Wang, Jialei Ji

**Affiliations:** 1Key Laboratory of Biology and Genetic Improvement of Horticultural Crops, Ministry of Agriculture, Institute of Vegetables and Flowers, Chinese Academy of Agricultural Sciences, Beijing 100081, China; caowenxue2018@163.com (W.C.); wp17356533891@163.com (P.W.); yanglimei@caas.cn (L.Y.); fangzhiyuan@caas.cn (Z.F.); zhangyangyong@caas.cn (Y.Z.); zhuangmu@caas.cn (M.Z.); lvhonghao@caas.cn (H.L.); wangyong03@caas.cn (Y.W.); 2College of Horticulture and Landscape Architecture, Hunan Agricultural University, 1 Nongda Road, Changsha 410128, China

**Keywords:** carotenoid biosynthetic genes, comparative genomics, *Brassica oleracea*, *PSY* (phytoene synthase) genes

## Abstract

Carotenoids are natural functional pigments produced by plants and microorganisms and play essential roles in human health. Cabbage (*Brassica oleracea* L. var. *capitata* L.) is an economically important vegetable in terms of production and consumption. It is highly nutritious and contains β-carotene, lutein, and other antioxidant carotenoids. Here, we systematically analyzed carotenoid biosynthetic genes (CBGs) on the whole genome to understand the carotenoid biosynthetic pathway in cabbage. In total, 62 CBGs were identified in the cabbage genome, which are orthologs of 47 CBGs in *Arabidopsis thaliana*. Out of the 62 CBGs, 46 genes in cabbage were mapped to nine chromosomes. Evolutionary analysis of carotenoid biosynthetic orthologous gene pairs among *B. oleracea*, *B. rapa*, and *A. thaliana* revealed that orthologous genes of *B. oleracea* underwent a negative selection similar to that of *B. rapa*. Expression analysis of the CBGs showed functional differentiation of orthologous gene copies in *B. oleracea* and *B. rapa*. Exogenous phytohormone treatment suggested that ETH, ABA, and MeJA can promote some important CBGs expression in cabbage. Phylogenetic analysis showed that *BoPSYs* exhibit high conservatism. Subcellular localization analysis indicated that *BoPSYs* are located in the chloroplast. This study is the first to study carotenoid biosynthesis genes in cabbage and provides a basis for further research on carotenoid metabolic mechanisms in cabbage.

## 1. Introduction

Cabbage (*Brassica oleracea* L. var. *capitata* L., 2n = 18) is a member of the family Cruciferae and one of the most economically important vegetable crops cultivated worldwide [1]. Cabbage is rich in dietary fiber, vitamin B1 (VB1), vitamin B (VB2), calcium, and iron. Most importantly, cabbage also contains β-carotene, lutein, and other antioxidant carotenoids. It is one of the best vegetables recommended by the World Health Organization [2,3].

Carotenoids are naturally occurring pigments mainly found in plants, algae, and photosynthetic bacteria. Carotenoid pigments are mostly C40 lipophilic isoprenoids, comprising eight isoprene units joined in a head-to-tail fashion, and the central unit has a reverse connection [4]. The number of conjugated double bonds determines the color of carotenoids; the more the number of conjugated double bonds, the deeper the red color [5]. So far, over 800 carotenoids have been found in nature [6]. Carotenoids are divided into two main groups based on whether they contain oxygen or not: carotenes and xanthophylls. Carotenes are composed of carbon and hydrogen atoms and include phytoene, lycopene, α-carotene, and β-carotene. On the other hand, xanthophylls are oxygenated hydrocarbons and include lutein, zeaxanthin, antheraxanthin, and violaxanthin [7,8]. Carotenoids are one of the components of the photosystem elements. They absorb light in the blue range of the spectrum and then transfer energy to chlorophylls in antenna complexes [4]. In plants, carotenoids protect proteins from excessive light incidence via thermal dissipation and free radical detoxification [9,10]. Carotenoids also act as vitamin A precursors and antioxidants, which play a vital role in human health and nutrition [11,12]. β-carotene is provitamin A, and its deficiency in a diet causes symptoms, such as night-blindness, keratomalacia, and xerophthalmia in humans [13]. Vitamin A deficiency also exacerbates afflictions like diarrhea, measles, and respiratory diseases [14,15]. α-carotene and β-carotene also serve as antioxidants in human health. For example, studies have shown that supplementing the human diet with carotenoids can reduce the risk of developing macular degeneration and neurodegenerative diseases [10,16].

Carotenoids are mainly synthesized from isopentenyl diphosphate (IPP) and dimethylallyl diphosphate (DMAPP) produced by the plastidic methylerythritol 4-phosphate (MEP) pathway [17,18] (Appendix A). The MEP pathway is localized in plastids and uses glyceraldehyde 3-phosphate and pyruvate as initial substrates to form deoxy-D-xylulose 5-phosphate (DXP). Subsequently, IPP and DMAP are generated by DXP synthase (DXS) and DXP reductoisomerase (DXR) and undergo a sequential series of condensation reactions to produce geranylgeranyl diphosphate (GGPP) [19]. Two GGPP molecules are condensed to phytoene by phytoene synthase (PSY). This is the first rate-limiting step of carotenoid biosynthesis, and PSY is the rate-limiting enzyme in this process [20,21]. Phytoene is further converted to lycopene by phytoene desaturase (PDS) and ζ-carotene desaturase (ZDS) [22,23]. Since carotenoids mainly exist in nature as trans structures, isomerization is key in carotenoid biosynthesis. Carotenoid isomerase (CRTISO) converts tetra-cis-lycopene to all-trans-lycopene [24,25,26]. The carotenoid pathway branches at the cyclization of lycopene, generating diverse carotenoids distinguished by different cyclic end groups. α-carotene and β-carotene are produced by lycopene cyclase (ε-LCY and β-LCY), catalyzing all-trans-lycopene [27,28], and then further hydroxylated to produce xanthophylls, including lutein and zeaxanthin [29]. Antheraxanthin and violaxanthin are sequentially synthesized by zeaxanthin epoxidase (ZEP). Violaxanthin is converted to neoxanthin by neoxanthin synthase (NSY). Neoxanthin can also be used to synthesize abscisic acid (ABA) [30,31].

Several transcription factors and genes participate in the regulatory network of carotenoid biosynthesis. Three *PSY* genes (*PSY1*, *PSY2*, and *PSY3*) were identified in tomato (*Solanum lycopersicum*). *PSY1* was mapped on chromosome 3 and is up-regulated in ripening tomato fruit. *PSY2* is specifically expressed in leaf tissues and is implicated in carotenoid synthesis in the leaves. *PSY3* is up-regulated in roots under stress conditions [32,33,34,35]. Additionally, several transcription factors directly or indirectly participate in carotenoid synthesis and metabolism. These transcription factors include MYB family, BBX family, NAC family, MADS-Box family, and AP2/ERF family among others [36,37,38,39,40]. Environmental factors such as light, temperature, and CO_2_ concentration also influence carotenoid synthesis to a great extent [9,41,42].

Although more in-depth studies on carotenoids synthesis have been carried out in *Arabidopsis thaliana* and tomato, there are few reports on the synthesis and metabolism of carotenoids in cabbage. In order to understand the mechanism of cabbage carotenoid synthesis more deeply and use modern molecular techniques to improve the carotenoids content in cabbage, we applied comparative genomics to identify cabbage carotenoids synthesis genes in the whole genome. Further, we determined the phylogenetic characteristics, chromosome location, non-synonymous/synonymous substitution (Ka/Ks), gene expression, and subcellular localization of genes related to carotenoid biosynthesis. The present study uncovers the genetic mechanisms underlying carotenoid biosynthesis in cabbage, thus laying a basis for future studies on carotenoid biosynthesis in plants.

## 2. Materials and Methods

### 2.1. Data Resource

The genomic and annotation data of *A. thaliana*, *B. oleracea*, and *Brassica rapa* were retrieved from the BRAD database (http://brassicadb.cn, accessed on 20 March 2021) and TAIR database (http://www.arabidopsis.org, accessed on 20 March 2021). The carotenoid biosynthetic gene (CBG) sequences of *A. thaliana* were acquired from the KEGG pathway database (http://www.genome.jp/kegg/pathway.html, accessed on 18 April 2021).

### 2.2. Identification and Analysis of Orthologs between B. oleracea and A. thaliana

To identify the CBGs and protein sequences of cabbage, we performed BLAST search on the cabbage genome and proteome database using *A. thaliana* carotenoid biosynthetic genes and protein sequences, with a cutoff E-value ≤ 10^−10^ and coverage ≥ 0.75. Syntenic orthologous genes between *A. thaliana* and *B. oleracea* were identified based on the collinearity of flanking genes sequence similarity (E-value ≤ 10^−20^). The specific distribution of *B. oleracea* carotenoid biosynthetic genes (BoCBGs) on the chromosome was analyzed using MapChart software (http://mapinspect.software.informer.com/, accessed on 24 April 2021).

### 2.3. Non-Synonymous/Synonymous Substitution (Ka/Ks) Ratios of Gene Pairs among B. oleracea, A. thaliana, and B. rapa

PAML software [43] was used to calculate the ratio of Ka/Ks rates of all orthologous gene pairs to estimate the selection mode of CBGs between *B. oleracea*, *A. thaliana*, and *B. rapa*. Ka/Ks ratio less than 1 indicates pure selection, ratio equal to 1 represents neutral selection, and ratio greater than 1 indicates positive selection.

### 2.4. Expression Analysis of Carotenoid Biosynthetic Genes in B. oleracea

The expression pattern of BoCBGs was investigated using RNA-Seq data [44]. Seven tissues, including root, leaf, stem, flower, bud, callus, and silique of *B. oleracea* accession ‘02-12’ were used for RNA extraction. Total RNA was isolated from the cabbage tissues using TIANGEN RNAprep Pure Plant Kit and then reverse-transcribed to cDNA using PrimeScript™ RT reagent kit (TaKaRa, Kyoto, Japan), according to the manufacturer’s instructions.

### 2.5. Exogenous Phytohormone Spraying of Cabbage Leaves

The leaves of cabbage were sprayed with different phytohormones to explore their effect on CBGs expression. Four exogenous phytohormones: ETH, ABA, SA, and MeJA at concentrations of 100 mg/L, 50 mg/L, 100 μM, and 100 μM, respectively, were sprayed. The leaves were sampled at 2, 4, 6, 12, and 24 h after spraying. The control was sprayed with double-distilled water alone. The relative expressions of *BoPSY.1*, *BoPDS.1*, *BoZDS*, and *BoLYC* genes were determined using qRT-PCR. The primers for qRT-PCR are listed in Appendix A.

### 2.6. Phylogenetic Analysis

PSY protein sequences from 24 plant species were analyzed. MEGA6.0 was used to construct the phylogenetic tree, and Poisson correction model was used for distance computation. Node support was assessed by 1000 bootstrap replicates.

### 2.7. Subcellular Localization of PSY in B. oleracea

The coding sequences of *BoPSY.1* (*Bol019820*), *BoPSY.2* (*Bol034439*), and *BoPSY.3* (*Bol021326*) were amplified using primers PSY1-1, PSY1-2, PSY2, and PSY3 (Appendix A), and then inserted into a pBWA(V)HS-GFP vector (BIORUN Technologies Company, Wuhan, China). The fusion constructs were introduced into tobacco leaf epidermis as previously described [45]. The fluorescence signals were observed using Olympus FV3000 confocal laser-scanning microscope.

## 3. Results

### 3.1. Identification of Carotenoid Biosynthetic Genes in B. oleracea

Genes associated with the carotenoid synthesis pathway in *A. thaliana* were analyzed using KEGG pathway database and TAIR database. A total of 47 genes were implicated in carotenoid biosynthesis in *A. thaliana*, of which 21 genes were shown to participate in the MEP pathway and 26 gene-encoding carotenoid biosynthetic enzymes (Table 1). A total of 62 CBGs were identified in cabbage, and 9 AtCBGs (*GGPS4*, *GGPS7*, *GGPS6*, *GGPS8*, *GGPS9*, *GGPS10*, *GGPS11*, *GGPS12*, and *LUT1*) showed no cabbage orthologs. Among the 62 BoCBGs, 58 were syntenic orthologs of the AtCBGs, and 4 BoCBGs had no syntenic relationships (Figure 1).

In this study, BoCBGs were divided into three sub-genomes (LF, MF1, and MF2). Among the 58 syntenic homologous genes, there are 27, 17, and 14 genes in LF, MF1, and MF2, respectively (Table 1). In addition, a collinearity relationship was observed between *B. oleracea* and *B. rapa*. In total, 43 pairs of carotenoid biosynthetic genes were mapped on chromosomes using Circos software (Figure 1).

### 3.2. Genomic Distribution on Chromosomes

Based on the genome annotation file, 62 BoCBGs were mapped to nine chromosomes. Among them, 46 were located on the chromosomes and 16 on the scaffold. The distribution of the 46 genes mapped on the chromosomes is shown in Figure 2, with 6, 3, 10, 4, 7, 2, 5, 6, and 3 genes located on chromosome C01–C09, respectively.

### 3.3. Comparative Evolutionary Analyses of Orthologous Gene Pairs for Carotenoid Biosynthetic Genes

Ka/Ks value is the ratio of non-synonymous mutation (Ka) to synonymous mutation (Ks) of genes. In this study, PAML software [43] was used to calculate the Ka/Ks of homologous gene pairs to explore the evolutionary selection mode of CBGs in *A. thaliana*, *B. oleracea* and *B. rapa*. Among the BoCBGs or BrCBGs, there were three copies of *AtPSY* gene, and two copies of *AtPDS3*, *AtLUT2*, and *AtCHY1* genes. Eventually, 20 orthologous gene pairs were selected from *A. thaliana*, *B. oleracea*, and *B. rapa* (Figure 3). Relative to *A. thaliana*, the Ka/Ks values of carotenoid homologous genes in *B. oleracea* and *B. rapa* were roughly the same, although the Ka/Ks ratios of *BoGGPS2*, *BoCHY1.1*, *BoCHY1.2*, and *BoCHY2.1* in *B. oleracea* genome were slightly higher than that of *B. rapa* genome. Also, *BoLUT2.1* and *BoLUT5.1* were lower than those in *B. rapa*. The Ka/Ks values shown in Figure 3 are all less than 1, indicating that these genes are subjected to purify selection in evolution.

### 3.4. Expression Analysis of Orthologous for Carotenoid Biosynthetic Genes among B. oleracea, B. rapa and A. thaliana

In this study, 20 CBGs in *A. thaliana* were studied to explore the differential expression of homologous genes in *B. oleracea* and *B. rapa* genomes. Four genes (*BoPSY.1*, *BoPSY.2*, *BrPSY1*, and *BrPSY2)* were homologous to *At5G17230* gene. *BoPSY.1* was expressed in all parts of *B. oleracea*, except roots, while *BoPSY.2* was expressed only in flowers and callus (Figure 4). *BrPSY1* was expressed in stems, leaves, flowers, and silique of *B. rapa*, while *BrPSY2* was almost exclusively expressed in flowers. The *BoPDS3.1* gene in *B. oleracea* was homologous to *At4G14210* and expressed in all tissues, but the *BrPDS3.1* was only expressed in silique of *B. rapa*. *At5G57030* exist in two copies in *B. oleracea* and *B. rapa*. *BoLUT2.1* expression was detected in all parts except roots, while *BrLUT2.1* expression was only observed in flowers and silique. *BoZEP.1*, *BoZEP.1*, *BrZEP1*, and *BrZEP2* are two copies of *At5G67030*. The expression level of *BoZEP.1* was highest in silique and the lowest in leaves, while that of *BrZEP1* exhibited an opposite trend. *BoNCED3.2* was expressed in all tissues of *B. oleracea*, but *BrNCED3.2* was detected in the callus of *B. rapa*. Analysis of differential expression between carotenoid homologous genes revealed functional variation among the homologous genes duplicated in *B. oleracea* and *B. rapa* during evolution.

The tissue expression of important genes in the carotenoid pathway was analyzed to further understand the expression of CBGs in *B. oleracea* (Figure 5). The results showed that *BoGGPS1* was expressed in all organs, including roots, leaves, stems, flowers, buds, silique, and callus, while *BoGGPS2* was only expressed in flower buds. The expression of *BoPSY.1* was highest in callus and flower buds, followed by flowers, stems, leaves, and silique. Almost no expression of *BoPSY.1* was detected in the roots. Meanwhile, the expression of *BoPSY.2* was highest in flowers, followed by buds and callus, and almost no expression was detected in other tissues. *BoPSY.3* was expressed in all tissues, with the highest expression detected in the stem. On the other hand, *BoPDS3.1* and *BoPDS3.2* were expressed in all tissues, however, *BoPDS3.1* was mainly expressed in callus, while *BoPDS3.2* was mainly expressed in the flowers. The expression of *BoZDS* was highest in the roots, while that of *BoLYC* was highest in the flowers. Gene expression analysis revealed an extensive variance between paralogs of each CBG.

### 3.5. BoPSY.1, BoPDS.1, BoZDS, and BoLYC Respond to Exogenous Phytohormone Treatments

Specific concentrations of ETH, ABA, SA, and MeJA were sprayed on cabbage leaves at the seedling stage to explore their effects on CBGs (*BoPSY.1*, *BoPDS.1*, *BoZDS*, and *BoLYC*). ETH at 100 mg/L promoted the expression of the four genes (Figure 6). Specifically, the expression of *BoPSY.1* increased at 2 h after treatment and was the highest at 12 h. Meanwhile, the expressions of *BoPDS.1*, *BoZDS*, and *BoLYC* were highest at 4 and 6 h and gradually decreased at 12 and 24 h. ABA also promoted the carotenoid pathway in cabbage. After spraying 50 mg/L ABA, the expression of *BoPSY.1* increased gradually and peaked after 24 h, while the expression of *BoZDS* increased first and then decreased. SA at 100 μmol/L promoted the expression of *BoPSY.1*, but had minimal effect on the other three genes. MeJA at 100 μmol/L increased the expressions of *BoPSY.1* and *BoPDS.1* to a certain extent, which then decreased and peaked after 24 h. Meanwhile, the expression of *BoZDS* gene peaked after 2 h of MeJA treatment. Collectively, these results indicate that ETH, ABA, and MeJA can promote CBGs expression in cabbage.

### 3.6. Evolution of PSY Genes

The protein sequences of *PSY* in 24 species were analyzed using MEGA 6.0 software to investigate the evolutionary relationship (Figure 7 and Appendix A). As shown in Figure 7, the phylogenetic tree was consistent with the taxonomic conclusion, indicating that *PSY* has a high conservatism, which provides a theoretical basis for plant genetic molecular evolution. The PSY proteins of 24 species were divided into two categories. *Haematococcus* clustered into one category alone. The protein cluster of the other plant *PSY* belonged to angiosperms and was further divided into five subgroups: rice, maize, and sorghum were grouped under *Commelinidae*; alfalfa, carrot, strawberry, and apple belonged to *Rosidae*; lilium brownii and narcissus belonged to the *Liliidae*; *Dilleniidae* was divided into two groups, one group includes Arabidopsis and cabbage (*Cruciferae*), papaya (*Carica Linn*), pumpkin, and watermelon (*Cucurbitaceae*), the other group includes persimmon and kiwifruit clustered with *Asteridae; Asteridae* includes tomato, chilli, medlar, sweet potato, cape jasmine, Osmanthus, and agastache. Notably, *BoPSY.1* and *BoPSY.3* clustered on one branch and were more closely related to *AtPSY* than *BoPSY.2* (Figure 7). Motif-based sequence analysis showed that all PSY proteins derived from angiosperms contain six conserved motifs, but the *Haematococcus* PSY only contains four motifs, lacking motif3 and motif6.

### 3.7. Subcellular Localization of BoPSY.1, BoPSY.2, and BoPSY.3

The results of Plant-PLoc subcellular localization prediction indicated that BoPSY proteins are localized in the chloroplast. Subcellular location of *BoPSYs* in tobacco (*Nicotiana benthamiana* L.) was examined using Confocal Laser Scanning Microscopy (FV3000). All three fluorescence signals of *BoPSY*-GFP fusion vector (*BoPSY.1*-GFP, *BoPSY.2*-GFP, and *BoPSY.3*-GFP) were observed in the chloroplast, consistent with the prediction of subcellular localization (Figure 8).

## 4. Discussion

### 4.1. Characterization of Carotenoid Biosynthetic Genes in B. oleracea

Whole genome duplication provides rich genetic material for the expansion of gene families or the evolution of new genes in plants [46,47]. A whole-genome triplication (WGT) event occurred in *B. oleracea* after its divergence from *A. thaliana* [44]. There are 47 genes associated with carotenoid synthesis in *Arabidopsis*. In this study, 62 CBGs were identified in cabbage, among which 58 were syntenic orthologs of *Arabidopsis*, and only four had no syntenic relationships, suggesting that they could have evolved from some duplicate genes of *Arabidopsis*. However, nine AtCBGs (*GGPS4*, *GGPS7*, *GGPS6*, *GGPS8*, *GGPS9*, *GGPS10*, *GGPS11*, *GGPS12*, and *LUT1*) showed no *B. oleracea* orthologs, indicating that they might have been lost during evolution of *B. oleracea*. Notably, *B. oleracea* and *A. thaliana* belong to the Brassicaceae family. Phylogenetic analysis revealed that the triplicated *B. oleracea* genome segments diverged from a common ancestor soon after the divergence of the *A. thaliana* and *Brassica* lineages [48,49]. The multiple copies of the BoCBGs that are syntenic to genes in *A. thaliana* were generated from the WGT. The fragment with the highest gene densities was in the sub-genome LF, the fragment with moderate gene densities was in the sub-genome MF1, and the fragment with the least genes was in the sub-genome MF2. There were 27, 17, and 14 genes located in LF, MF1, and MF2, respectively. The distribution of CBGs was in line with the gene separation status in the whole genomic sequences [44].

Further, we compared the non-synonymous/synonymous (Ka/Ks) ratio of orthologous gene pairs between *A. thaliana*, *B. oleracea*, and *B. rapa* to analyze the evolutionary relationship of orthologous gene pairs for CBGs among the three species (Figure 3). The Ka/Ks ratio is a measure of selective pressures acting on genes. There were no significant differences in orthologous gene pairs between *A. thaliana-B. oleracea* and *A. thaliana-B. rapa* lineages. Therefore, we speculated that BoCBGs and BrCBGs could have undergone similar negative selection. However, comparing the expression differences between orthologous gene pairs in CBG revealed functional variation of a few gene pairs in *B. oleracea* and *B. rapa*.

### 4.2. Exogenous Phytohormones Regulate the Expression of Carotenoid Biosynthetic Genes

Regulation of carotenoid biosynthesis in plants is a complex process that is regulated by multilevel factors. Exogenous phytohormones affect the accumulation of carotenoids. Ethylene (ET) plays a vital role in plant ripening and carotenoid accumulation. Thus, inhibiting ET synthesis in fruits can hinder fruit ripening and lycopene accumulation [50]. Abscisic acid (ABA) is one of the most important plant phytohormones. Its synthetic precursors include neoxanthin and violaxanthin. These precursors regulate ABA synthesis, which in turn mediate the synthesis of carotenoids in plants [51,52]. Salicylic acid (SA) and Methyl Jasmonate (MeJA) also play an essential role in plant growth and stress resistance.

In the present study, different concentrations of exogenous phytohormones (ETH, ABA, SA, and MeJA) were sprayed on cabbage leaves to explore their effects on CBGs expression. The expression of four CBGs (*BoPSY.1*, *BoPDS.1*, *BoZDS*, and *BoLYC*) was analyzed. The result showed that ETH, ABA, and MeJA could promote the expression of CBGs in cabbage at specific concentrations. SA at 100 μmol/L did not significantly affect the expression of the four genes, consistent with the report of Gao, who concluded that the carotenoid synthesis pathway is insensitive to SA [53]. These results will guide the adoption of exogenous phytohormone spraying to increase carotenoid contents in cabbage.

### 4.3. Phytoene Synthase (PSY) Is a Key Enzyme in the Carotenoid Biosynthetic Pathway

PSY is the first special enzyme in the carotenoid biosynthesis pathway, catalyzing the condensation of two GGPP molecules into phytoene [54]. Overexpression of *PSY* increased carotenoid content and substantially improved β-carotene synthesis in canola seeds, cassava roots, potato tubers, and endosperms [55,56,57]. In tomato, *PSY1* is highly expressed in fruits, *PSY2* is essential for carotenoid synthesis in leaf tissues, and *PSY3* potentially functions in roots [32,33,34,35].

In this study, *BoPSY.1* and *BoPSY.3* clustered on one branch, with *BoPSY.2* in a separate branch. However, the three *BoPSYs* clustered on a specific branch in *Arabidopsis*, where a single *PSY* gene was shown to regulate phytoene synthesis in all tissues [21]. Three genes encoding *PSY* enzymes were identified in *B. oleracea* and *B. rapa* after WGT event [58]. In this study, three *BoPSYs* genes in cabbage were all localized in the chloroplast, which is similar to maize *PSY1* [59]. However, examination of tissue expression of *BoPSYs* showed that they are differentially expressed in cabbage tissues (Figure 5). Thus, we speculate that the *BoPSYs* might function in different cabbage tissues, like the *PSYs* expression patterns in tomato [32,33,34,35]; however, further studies are needed to verify this hypothesis.

## 5. Conclusions

In total, 62 carotenoid biosynthetic genes were identified in cabbage. These genes in *B. oleracea* species underwent similar negative selection with those in *B. rapa* species. However, the differential expression pattern of the duplicated CBGs occurred after polyploidization. The expression of BoCBGs can be regulated by phytohormones. ETH, ABA, and MeJA can promote the expression of *BoPSY.1*, *BoPDS.1*, *BoZDS*, and *BoLYC*, which provides a basis for spraying phytohormones to improve the carotenoid content of cabbage. As the key enzyme in the carotenoid biosynthetic pathway, three BoPSY proteins were all localized in chloroplasts and exhibited different expression patterns in various cabbage tissues. This study uncovered carotenoid metabolic mechanisms in *B. oleracea*, proving a basis for developing cabbage varieties with high carotenoid content via genetic engineering.

## Figures and Tables

**Figure 1 genes-12-02027-f001:**
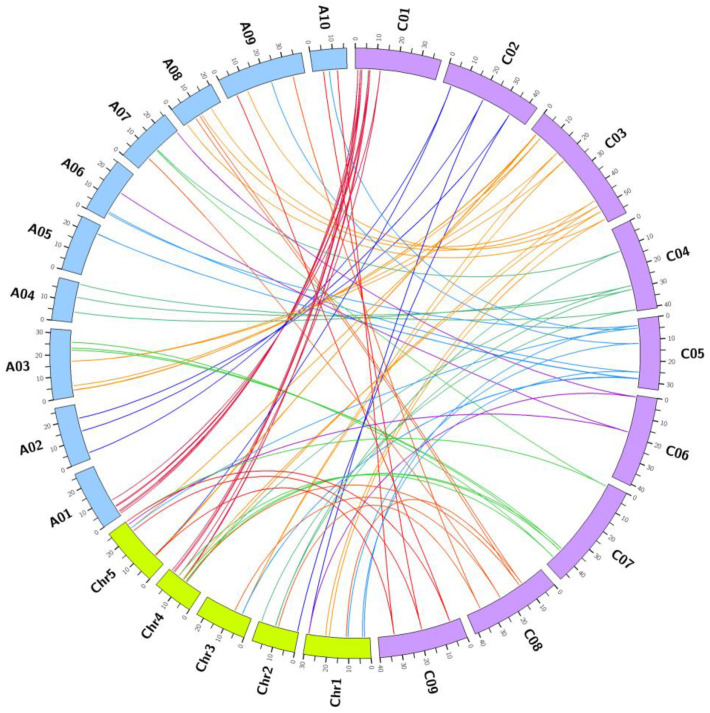
Circos diagram of syntenic carotenoid biosynthetic genes in *B. oleracea*, *B. rapa*, and *A. thaliana*. C01 to C09 indicate *B. oleracea* chromosomes, A01 to A10 indicate *B. rapa* chromosomes, and Chr1 to Chr5 indicate *A. thaliana* chromosomes.

**Figure 2 genes-12-02027-f002:**
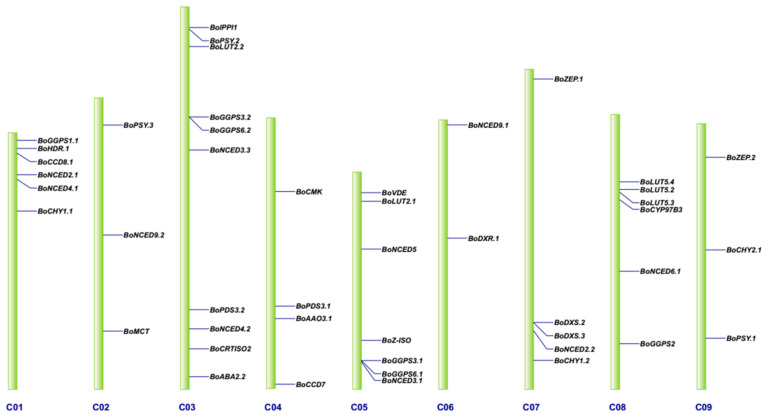
Genomic localization of the carotenoid biosynthetic genes on the nine chromosomes of *B. oleracea*. C01 to C09 indicate *B. oleracea* chromosomes.

**Figure 3 genes-12-02027-f003:**
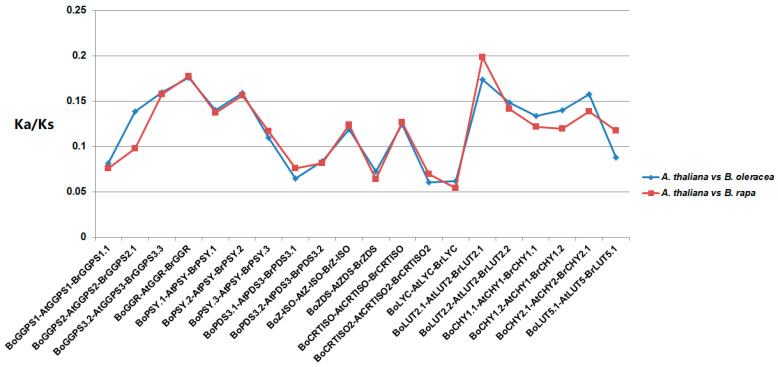
Ka/Ks values of orthologous gene pairs for carotenoid biosynthetic genes among *A. thaliana*, *B. oleracea* and *B. rapa*.

**Figure 4 genes-12-02027-f004:**
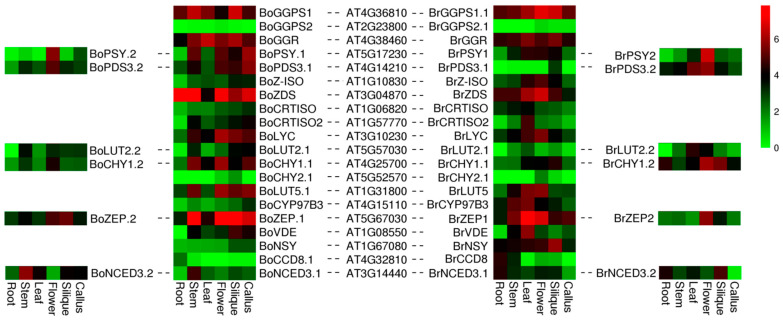
Heat map representation of orthologous gene pairs of carotenoid biosynthetic genes of *B. oleracea* and *B. rapa*. Color scale bar represents log_2_ transformed FPKM values.

**Figure 5 genes-12-02027-f005:**
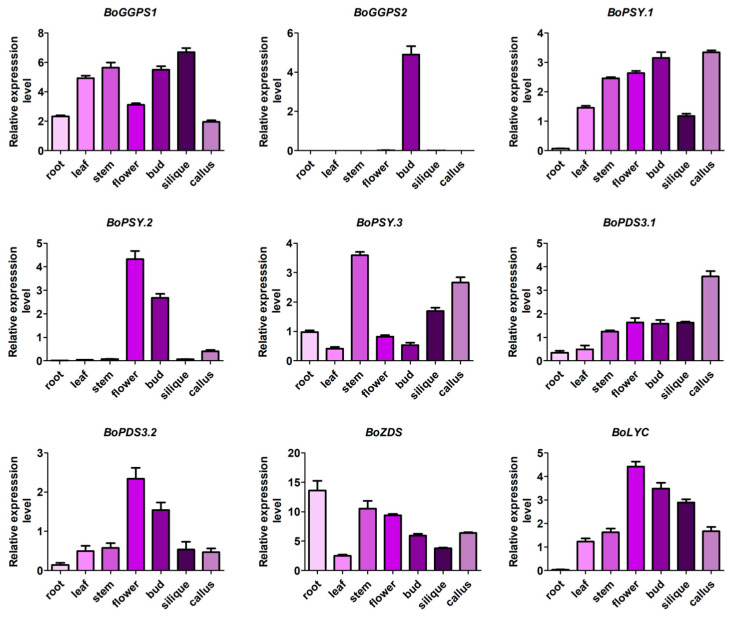
Relative expression levels of nine key carotenoid biosynthetic genes in different tissues of cabbage.

**Figure 6 genes-12-02027-f006:**
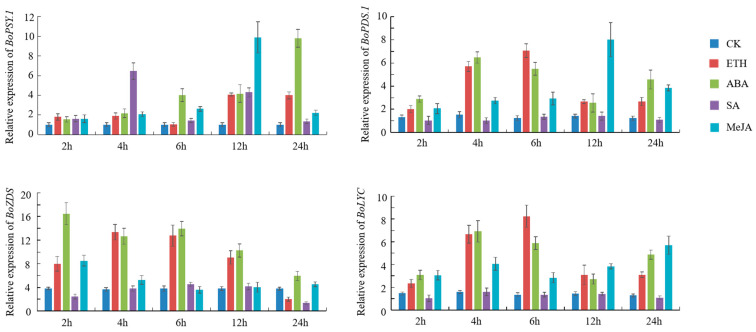
The expression of *BoPSY.1*, *BoPDS.1*, *BoZDS*, and *BoLYC* after treatment with exogenous phytohormone. The *Y*-axis and *X*-axis represent the relative expression level and the time course of exogenous phytohormone treatment, respectively. Leaves were sampled at 0, 2, 4, 6, 12, and 24 h after ETH, ABA, SA, and MeJA spraying. Data represent the mean ± SD of three technical repetitions.

**Figure 7 genes-12-02027-f007:**
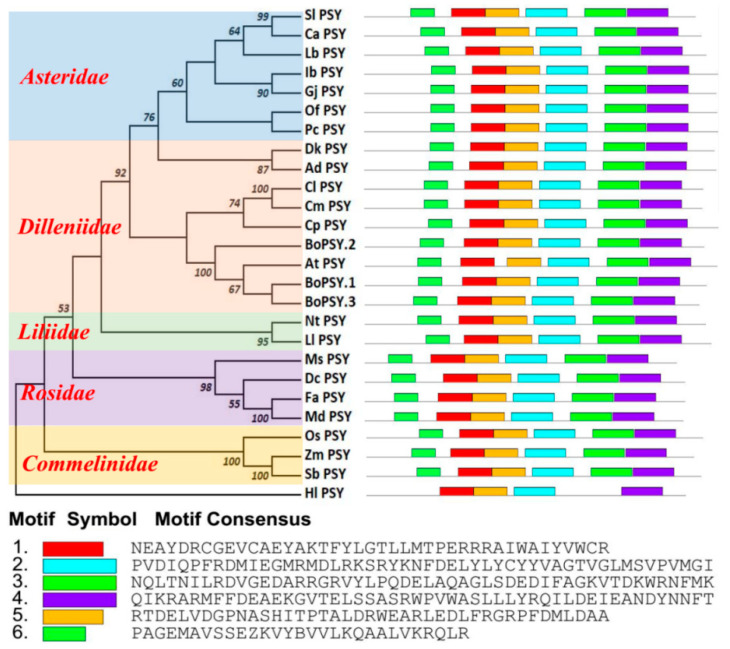
Phylogenetic relationship and conserved motif analysis of *PSY* genes. The phylogenetic tree was generated using the Muscle module and neighbor-joining (NJ) method with 1000 bootstrap replicates implemented in MEGA 6.0. Sl: *Solanum lycopersicum*; Ca: *Capsicum annuum*; Lb: *Lycium barbarum*; Ib: *Ipomoea batatas*; Gj: *Gardenia jasminoides*; Of: *Osmanthus fragrans*; Pc: *Pogostemon cablin*; Dk: *Diospyros kaki*; Ad: *Actinidia deliciosa*; Cl: *Citrullus lanatus*; Cm: *Cucurbita moschata*; Cp: *Carica papaya*; Bo: *Brassica oleracea*; At: *Arabidopsis thaliala*; Nt: *Narcissus tazetta;* Ll: *Lilium lancifolium;* Ms: *Medicago sativa*; Dc: *Daucus carota*; Fa: *Fragaria ananassa*; Md: *Malus domestica*; Os: *Oryza sativa Indica*; Zm: *Zea mays*; Sb: *Sorghum bicolor*; Hl: *Haematococcus lacustris*.

**Figure 8 genes-12-02027-f008:**
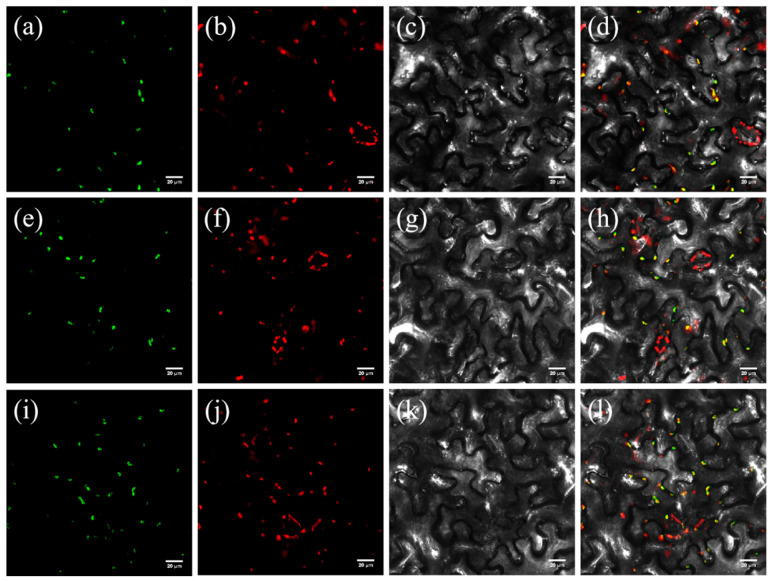
Subcellular localization of BoPSY proteins in tobacco leaf epidermis cells. (**a**–**d**) Subcellular localization of BoPSY.1. (**e**–**h**) Subcellular localization of BoPSY.2. (**i**–**l**) Subcellular localization of BoPSY.3. (**a**,**e**,**i**) Fluorescence signals of BoPSYs fused with GFP. (**b**,**f**,**j**) Fluorescence signal of chloroplast. (**c**,**g**,**k**) Bright field images. (**d**,**h**,**l**) Superposition images of bright field and fluorescence.

**Table 1 genes-12-02027-t001:** Carotenoid biosynthetic genes identified in cabbage.

Enzyme	*Arabidopsis thaliana*	*B. olerecea*			
		Syntenic Orthologs			Non-Syntenic Orthologs
		LF	MF1	MF2	
*DXS*	*AT4G15560*	*BoDXS.1 (Bol020307)*	*BoDXS.2 (Bol005061)*	-	*BoDXS.3 (Bol005062)*
*DXR*	*AT5G62790*	*BoDXR.1 (Bol019430)*	-	*BoDXR.2 (Bol019181)*	-
*MCT*	*AT2G02500*	-	*BoMCT (Bol015155)*	-	-
*CMK*	*AT2G26930*	*BoCMK (Bol027780)*	-	-	-
*MDS*	*AT1G63970*	*BoMDS.1 (Bol021045)*	-	*BoMDS.2 (Bol022634)*	-
*HDS*	*AT5G60600*	*BoHDS (Bol003103)*	-	-	-
*HDR*	*AT4G34350*	*BoHDR.1 (Bol013657)*	-	*BoHDR.2 (Bol001484)*	-
*IPPI1*	*AT5G16440*	-	*BoIPPI1 (Bol034402)*	-	-
*IPPI2*	*AT3G02780*	*BoIPPI2.1 (Bol000950)*	*BoIPPI2.2 (Bol001738)*	*BoIPPI2.3 (Bol002232)*	-
*GGPS1*	*AT4G36810*	*BoGGPS1 (Bol028967)*	-	-	-
*GGPS2*	*AT2G23800*	*BoGGPS2 (Bol045796)*	-	-	-
*GGPS3*	*AT3G14550*	*BoGGPS3.1 (Bol005099)*	-	*BoGGPS3.2 (Bol025714)*	-
*GGPS7*	*AT2G18620*	-	-	-	-
*GGPS8*	*AT3G14510*	-	-	-	-
*GGPS11*	*AT3G29430*	-	-	-	-
*GGPS4*	*AT2G18640*	-	-	-	-
*GGPS6*	*AT1G49530*	-	-	-	-
*GGPS9*	*AT3G14530*	-	-	-	-
*GGPS10*	*AT3G20160*	-	-	-	-
*GGPS12*	*AT3G32040*	-	-	-	-
*GGR*	*AT4G38460*	*BoGGR (Bol000792)*	-	-	-
*PSY*	*AT5G17230*	*BoPSY.1 (Bol019820)*	*BoPSY.2 (Bol034439)*	*BoPSY.3 (Bol021326)*	-
*PDS3*	*AT4G14210*	-	*BoPDS3.1 (Bol009962)*	*BoPDS3.2 (Bol016089)*	-
*Z-ISO*	*AT1G10830*	*BoZ-ISO (Bol036691)*	-	-	-
*ZDS*	*AT3G04870*	-	*BoZDS (Bol003131)*	-	-
*CRTISO*	*AT1G06820*	-	-	*BoCRTISO (Bol004146)*	-
*CRTISO2*	*AT1G57770*	-	-	*BoCRTISO2 (Bol016311)*	-
*LYC*	*AT3G10230*	*BoLYC (Bol011368)*	-	-	-
*LUT2*	*AT5G57030*	*BoLUT2.1 (Bol037991)*	*BoLUT2.2 (Bol026053)*	-	-
*CHY1*	*AT4G25700*	*BoCHY1.1 (Bol039555)*	*BoCHY1.2 (Bol042254)*	-	-
*CHY2*	*AT5G52570*	*BoCHY2.1 (Bol030235)*	-	*BoCHY2.2 (Bol045243)*	-
*LUT5*	*AT1G31800*	-	*BoLUT5.1 (Bol027064)*	-	*BoLUT5.2 (Bol027080)* *BoLUT5.3 (Bol014196)*
*CYP97B3*	*AT4G15110*	-	-	*BoCYP97B3 (Bol028312)*	-
*LUT1*	*AT3G53130*	-	-	-	-
*ZEP*	*AT5G67030*	*BoZEP.1 (Bol027222)*	-	*BoZEP.2 (Bol019241)*	-
*VDE*	*AT1G08550*	*BoVDE (Bol041221)*	-	-	-
*NSY*	*AT1G67080*	-	*BoNSY (Bol045096)*	-	-
*CCD7*	*AT2G44990*	-	*BoCCD7 (Bol021707)*	-	-
*CCD8*	*AT4G32810*	*BoCCD8.1 (Bol017864)*	-	-	*BoCCD8.2 (Bol000605)*
*NCED2*	*AT4G18350*	*BoNCED2.1 (Bol009433)*	*BoNCED2.2 (Bol037062)*	-	-
*NCED3*	*AT3G14440*	*BoNCED3.1 (Bol005093)*	*BoNCED3.2 (Bol011830)*	*BoNCED3.3 (Bol035582)*	-
*NCED4*	*AT4G19170*	*BoNCED4.1 (Bol009345)*	-	*BoNCED4.2 (Bol029878)*	-
*NCED5*	*AT1G30100*	*BoNCED5 (Bol022516)*	-	-	-
*NCED6*	*AT3G24220*	*BoNCED6 (Bol007451)*	-	-	-
*NCED9*	*AT1G78390*	*BoNCED9.1 (Bol027485)*	*BoNCED9.2 (Bol018961)*	-	-
*ABA2*	*AT1G52340*	*BoABA2.1 (Bol000478)*	*BoABA2.2 (Bol035060)*	-	-
*AAO3*	*AT2G27150*	-	*BoAAO3.1 (Bol026459)*	-	-

## Data Availability

Data will be available on reasonable request.

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
