# Peer review of "Carotenoid Biosynthetic Genes in Cabbage: Genome-Wide Identification, Evolution, and Expression Analysis"

_genes, 2021, doi:10.3390/genes12122027_

Round 1
Reviewer 1 Report
The manuscript details an analysis of carotenoid biosynthesis genes in cabbage. The data is valuable. So I can confirm that the subject matter of this paper is of interest and relevance for publication in Genes.
I recommend this paper – after minor revision.
Comment to the Authors:
- Brassica oleracea var. capitate or Brassica oleracea correct on: Brassica oleracea var. capitata L. - in manuscript
- the authors did not present any aim and hypothesis at the beginning
- in Conclusion - give few key message - an idea may be to synthetize in 3-5 bullet the key results of the study, evidences and recommendation. This improvement will increase clearness and readability. Add a practical implications statement.
Author Response
Point 1: Brassica oleracea var. capitate or Brassica oleracea correct on: Brassica oleracea var. capitata L. - in manuscript
Response 1: We are very sorry for our negligence of this point, we have changed the ‘Brassica oleracea var. capitate’ to ‘Brassica oleracea L. var. capitata L.’ (Line 15), and consider the brevity of the manuscript, ‘Brassica oleracea’ has been reserved.
Point 2: The authors did not present any aim and hypothesis at the beginning.
Response 2: Thank you for your nice suggestion. We have supplemented the aim and hypothesis (line 91-96).
Point 3: in Conclusion - give few key message - an idea may be to synthetize in 3-5 bullet the key results of the study, evidences and recommendation. This improvement will increase clearness and readability. Add a practical implications statement.
Response 3: Considering this suggestion, we have rewritten the Conclusion.
Reviewer 2 Report
In this paper, the authors identified the carotenoid biosynthetic genes and analyzed the expression of them in cabbage. The authors also discussed the evolution of them. These findings are useful for the readers of Genes.
Author Response
Point 1: In this paper, the authors identified the carotenoid biosynthetic genes and analyzed the expression of them in cabbage. The authors also discussed the evolution of them. These findings are useful for the readers of Genes.
Response 1: Thank you for your recommendation.